# Lumboperitoneal Shunt: A New Modified Surgical Technique and a Comparison of the Complications with Ventriculoperitoneal Shunt in a Single Center

**DOI:** 10.3390/medicina55100643

**Published:** 2019-09-26

**Authors:** Tsung-Hsi Yang, Cheng-Siu Chang, Wen-Wei Sung, Jung-Tung Liu

**Affiliations:** 1Department of Neurosurgery, Chung Shan Medical University Hospital, Taichung 402, Taiwan; jimmyandpipi@hotmail.com (T.-H.Y.); chengsiu.chang@gmail.com (C.-S.C.); flutewayne@gmail.com (W.-W.S.); 2Institute of Medicine, Chung Shan Medical University, Taichung 402, Taiwan; 3Department of Urology, Chung Shan Medical University Hospital, Taichung 402, Taiwan

**Keywords:** complication, lumboperitoneal shunt, surgical technique, ventriculoperitoneal shunt

## Abstract

*Background and objectives*: Hydrocephalus remains a disease requiring surgical treatment even in the modern era. Ventriculoperitoneal (VP) shunt placement is the most common treatment, whereas lumboperitoneal (LP) shunts are less commonly used due to initial reports of very high rates of complications. In the present study, we retrospectively reviewed our experience of the new two-stage procedure with LP shunt implantation to assess the complications and the results of this procedure versus VP shunt insertion. *Materials and Methods*: All patients from a single center who had received LP shunts using a Medtronic Strata device or VP shunts in the past six-year interval were retrospectively reviewed. The LP shunt insertion was a new two-stage procedure. We compared the three major complications and shunt revisions between the two groups, including shunt malfunction, infection, and subdural hematoma. *Results*: After matching the age and sex of both groups, we included 96 surgery numbers of LP shunts and 192 surgery numbers of VP shunts for comparison. In the LP shunt group, one patient (1.0%) underwent revision of the shunt due to shunt infection. In the VP shunt group, 26 surgeries (13.5%) needed revision, and 11 surgeries (5.7%) had shunt infection. Shunt malfunction occurred in 14 patients (7.3%) and all needed revisions. The revision rate showed statistically significant differences between the LP and VP shunt groups (*p* < 0.001). *Conclusions*: The recent improvements in the quality of the LP shunt device and the proficiency of the procedure has made the LP shunt a safer procedure than the VP shunt. The programmable valve can avoid overdrainage complications and reduce the revision rate. With our procedural steps, the LP shunt can be used to decrease the complications and revision rates.

## 1. Introduction

Hydrocephalus remains a disease requiring surgical treatment even today. The most common treatment is ventriculoperitoneal (VP) shunt placement, but this procedure has many complications, such as intraparenchymal or intraventricular hemorrhage, seizure, malposition, and infection [1]. An alternative procedure is the lumboperitoneal (LP) shunt, but this is used much less frequently because of initial reports of a very high occurrence of complications due to the lack of a reservoir to adjust the flow of cerebrospinal fluid (CSF), which has led to underutilization of the procedure. In recent years, the use of programmable valves has become more routine for reducing the complications of overdrainage, which can include subdural hematoma, intracerebral hemorrhage, or cerebellar tonsillar herniation [2].

The traditional surgical technique for LP shunt placement was a one-stage surgery with the patient in the lateral decubitus position [3,4]. However, it had many disadvantages, such as spinal catheter kinking, malposition, puncture trauma, and a higher infection and revision rate. In the present study, we modified the LP procedure to a two-stage surgery and standardized the application of our technique to each case. Our procedure is very easy to copy for all neurosurgeons and only needs an additional mobile C-arm X-ray system, which is easily obtained in the operating room. The details of our surgical technique are described in the methods. We also conducted a retrospective review of our experience with LP shunt implantation to assess the complications and results of this procedure, and to compare the complications of the LP shunt to that of the VP shunt.

## 2. Materials and Methods

### 2.1. Patients

The data for all patients who received LP shunts and VP shunts at Chung Shan Medical University Hospital, Taichung, Taiwan, from January 2012 to December 2017 were retrospectively reviewed (The Institutional Review Board: CS18155). In total, 136 patients receiving LP shunts and 299 patients receiving VP shunts were included (all patients were adults). After matching the age and sex of both groups, we included 96 surgery numbers of LP shunts and 192 surgery numbers of VP shunts for comparison. All surgeries were performed by the same senior neurosurgeon. The average follow-up period was 3.4 years. Every patient received brain magnetic resonance imaging (MRI) or computed tomography (CT) before the surgery and then follow-up image studies were performed 3 and 6 months after the operation for complication survey. Our indications for LP shunt included communicating hydrocephalus (the most common cause containing infections and post-hemorrhage), idiopathic intracranial hypertension, and normal pressure hydrocephalus. The indications for VP shunts were the same as for LP shunts.

### 2.2. Surgery Procedure

#### 2.2.1. Pre-Operative Assessment for LP Shunt Group

Brain MRI or CT was performed to confirm the hydrocephalus diagnosis. A whole spinal series MRI was also performed to confirm the patency of the spinal canal. We were able to identify a severe stenosis of the spinal canal or a spinal tumor that might block the patency of the CSF from the ventricle to LP shunt. The spine MRI was used to facilitate carrying out the tapping step while avoiding the stenosis segment (Figure 1). We used Medtronic Strata^®^ NSC LP adjustable pressure shunts for all patients and these were initially set at the highest pressure (pressure: 2.5, 20 cm H_2_O).

#### 2.2.2. Procedure of the New Two-Stage Procedure for LP Shunts Insertion

In contrast to the usual procedure, which uses a lateral decubitus position and a single-stage surgery, we use a two-stage surgery in which the patient was first in the prone and then in the supine position. In the first stage, with the patient in the prone position, a 2 cm skin incision was made and a Tuohy needle was inserted into the subarachnoid space at the L2–S1 level (according to the whole spine MRI) and then the lumbar catheter was inserted under fluoroscopic guidance (Figure 2). We were able to avoid the disadvantages of blind tapping and issues with spinal catheter insertion, such as subarachnoid hemorrhage and catheter kinking inside the thecal sac. The tube was then placed in a subcutaneous pocket made at the flank region (Figure 3). An anchor was used to hold the lumbar catheter and sutures were used to fix the subcutaneous tissue. In the secondary stage, with the patient in the supine position, we passed the tunneler from an abdominal incision to the flank region and fed the catheter from the flank to the abdomen. We then connected the catheter with a Strata adjustable valve and placed the abdominal catheter inside the peritoneum.

#### 2.2.3. Procedure of the VP Shunt Insertion

The patient is positioned supine with the head turned to the left (or right) and a bump is placed under the shoulders. We use an open technique for abdominal catheter insertion that was the same as the LP shunt groups. A vertical incision median and superior to the umbilicus was made and fascial layers were incised. The peritoneum was gently opened for visual confirmation of entry into the peritoneal cavity. A cranial incision is made in a curvilinear fashion over the right or left Kocher’s point of lateral ventricular catheter insertion. After creating a subcutaneous pocket to accommodate the valve, a subcutaneous tunneler was passed from the cranial to the abdominal incision. All the VP shunt catheters were connected with a Strata^®^ NSC adjustable pressure valve. After confirming the steady egress of CSF from abdominal catheter, the catheter was placed inside the peritoneum.

Here, we present a case of treatment with LP shunt implantation. This 67-year-old male presented with the clinical symptoms of dementia and gait disturbance. His symptoms progressed over time in one year. Brain CT showed hydrocephalus (Figure 4A). His clinical symptoms, duration, and image met the criteria of probable normal pressure hydrocephalus. The modified two-stage LP shunting procedures (Strata^®^ NSC LP valve, initial pressure setting: 2.5, 20 cm H_2_O) were performed to control the hydrocephalus. The pressure of the valve was adjusted once at 3 months post-brain CT (pressure 2.5, 20 cm H_2_O to 2.0, 14.5 cm H_2_O). The brain CT images at 3 and 6 months post-operation are shown in Figure 4B,C. The obvious decreased ventricle sizes and his symptoms improved gradually without any side effects at our clinic followed-up.

### 2.3. Statistical Analysis

We compared the three major complications (including shunt malfunction, infection, and subdural hematoma) and shunt revisions between the LP and VP shunt groups. The χ2 test and Fisher’s exact test were applied for continuous or discrete data analysis. All statistical analyses were conducted using SPSS statistical software (version 15.0; SPSS, Inc., Chicago, IL, USA). We considered *p* < 0.05 to indicate a statistically significant difference.

## 3. Results

In our study, in the past 6 years, 137 patients with hydrocephalus underwent 138 LP shunt implantations while 299 patients underwent 345 VP shunt operations. After matching the age and sex of both groups, we included 96 surgery numbers of LP shunts and 192 surgery numbers of VP shunts for comparison. In the LP shunt group, one patient (1.0%) underwent revision of the LP shunt due to shunt infection. One patient (1.0%) experienced chronic subdural hematoma and subdural effusion due to overshunting after adjustment of the valve to a lower pressure. In this case, the symptoms were improved by adjusting the Strata valve to a higher pressure, and a second operation was not necessary. No acute hemorrhage (including acute subdural hematoma, intraparenchymal hemorrhage, or intraventricular hemorrhage), seizure, acquired Chiari malformation, or shunt malfunction (including obstruction or migration of catheter) was observed.

In the VP shunt group, 26 surgeries (13.5%) needed revision. Of these, 11 (5.7%) surgeries had shunt infection (7 patients need revisions once and 2 patients underwent two revisions). Shunt malfunction occurred in patients (7.3%) and all needed revisions. Subdural hematoma occurred in 5 patients (2.6%) and one patient needed revision.

The revision rate was significantly larger for the VP group than for the LP group (*p* < 0.01) (Table 1). The malfunction rate was lower in the LP group than in the VP group, but the number for the LP group was zero so a *p* value could not be calculated. Then infection rate of LP group was lower than in the VP group, but was not considered significantly different (*p* = 0.061). The numbers of subdural hematomas did not differ significantly between the two groups (*p* = 0.766)

## 4. Discussion

The LP shunt has several advantages over the VP shunt, including fewer complications and the avoidance of any cranial surgery; the latter reduces the risk of ventricular puncture, which can cause intracerebral hemorrhage and seizure. The use of the LP shunt can also reduce the infection rate because bacteria are much less prevalent on the back skin than they are on the scalp hair follicles. In our study, the infection rate, revision rate, and shunt malfunction rate were significantly lower in LP shunt cases than in VP shunt cases. The same conclusion was made in the series by Aoki et al., Singh et al., and Duthel et al. [5,6,7]. Although the three authors performed traditional one-stage LP procedures with lateral decubitus position, the bacteria from hair follicles on the scalp were avoided such that we got the same results.

We also compared our data for LP shunts with the large number of VP shunts reported by Wu et al. (14,455 individuals who underwent VP shunt insertion with 65,040 person-years of follow-up), and we found significantly lower revision rates and few of any type of shunt complication in our LP group than in their VP group (revision rate: 0.7% versus 21.6%; any shunt complication: 2.0% versus 29%, *p* < 0.001) [8]. When compared with the infection rate for VP shunts reported by Reddy et al. (1015 patients), our LP shunt cases showed a significantly lower infection rate (0.7% versus 7.2%, *p* < 0.001) (Table 2) [9].

Chumas et al. revealed that LP shunts could be successfully used in the pediatric population [10]; however, our study population contained no pediatric cases. Duthel et al. reported that the indication for an LP shunt must be strictly discussed because of the frequency of general and special complications [7]. Singh et al. found that the non-obstructive complication rates were lower for the LP shunt than for the VP shunt in post-meningitis communicating hydrocephalus [6]. More data would seem to be needed to reach a consensus on the use of LP shunts.

Pre-operative whole spine MRI provides useful information of spinal canal condition, so this MRI procedure could increase the success rate of the LP shunt operation. Performing a spinal puncture is difficult for many reasons, including severe spinal canal stenosis, severe disc herniation, spinal tumor, interlaminar space stenosis, and spinal deformity. Even severe stenosis of the cervical spinal canal can decrease the flow of CSF from the brain to the LP shunt. Once we puncture at the stenosis level, this might increase the frequency of tapping, which can easily cause trauma or hematoma and increase the chance of irritation and radiculopathy due to contact of the spinal catheter with the rootlet. For example, if stenosis is at the L3–4 level, we should tap from the L2–3 level to decrease the possibility of complications.

In contrast to other one-stage surgeries performed in the lateral decubitus position, our modified two-stage surgical technique for LP shunt placement has several advantages:(1)It allowed us to easily perform lumbar tapping and insertion of the spinal catheter under fluoroscopic guidance. Sabah et al. revealed that insertion of an LP shunt without fluoroscopy guidance has a 15.8% chance of false positioning of the proximal end of the spinal catheter [11]. We designed the two-stage surgery with the patient in a prone then a supine position. In the first stage, the prone position simplified checking the true lateral view of the spine X-ray so that we could actually place the catheter in the correct position.(2)It ensured one lumbar tapping into intrathecal sac. This decreased the trauma rate during the puncture, as trauma might cause severe complications. Basaran et al. showed a case report of a spinal intradural hematoma and permanent paraparesis after an LP shunt insertion, while other studies have also reported the induction of epidural hematomas by spinal puncture [12,13,14].(3)It avoided the kinking of the lumbar spinal catheter in the thecal sac. In our experience, spinal catheter entanglement or twisting occurred in about 10%–15% cases, but we were able to correct it immediately under fluoroscopic guidance (Figure 5).(4)It ensured the level of the lumbar catheter tip placement. We placed the spinal catheter tip at the T8–10 level, which helped to avoid extraction of the spinal catheter from thecal sac when the patient performs strenuous exercise. Matsubara et al. revealed a case report of CSF leakage into the epidural space through a side hole, due to placement of a length of the spinal catheter into the dural sac. Shortly after the operation, the side hole was opened up when the patient moved his body [15].

Common complications of the LP shunt include shunt infection; malfunction (including dislocation, disconnection, migration, and obstruction); Arnold–Chiari I malformation (ACM); subdural hematoma due to overdrainage; CSF leakage; and radicular pain [16,17,18]. We compared the common complications with those reported in four series published from 1990 to 2012 (Table 3) [5,7,18,19]. The revision rate for our study was 0.7%, and was significantly lower than all previously published values (*p* < 0.001). One case of shunt infection was seen (0.7%); this incidence of infection was similar to that of the series reported by Aoki (1%) and was significantly lower than in other previously reported work. We considered that the reason for this discrepancy may be the two-stage surgery using two body positions. The prone and supine positions provide a flat operating field that was much easier to keep sterile when compared to the lateral decubitus position, where maintaining sterile conditions is difficult in heavy patients due to their many skinfolds.

The shunt malfunction rate varied from 12.1% to 14% in the four previously published series, but our rate was nil in the present study, which was apparently attributable to our modified surgical technique. The occurrence of subdural hematoma could not be completely avoided in all series, including ours, perhaps because of overdrainage and inappropriate pressure by the valve. The use of programmable valves for our all patients allowed recovery from these complications by adjusting to an appropriate pressure. The other complications, such as CSF leakage, ACM, and radicular pain, were nil in our patients.

The efficacy of improving the clinical symptoms of patients was not determined in our study. However, many previous studies have determined that LP shunts can be equally as effective as VP shunts for the treatment of hydrocephalus [19,20,21,22,23]. For example, Masakaze et al. considered that the efficacy and safety rates of LP shunts were comparable to those of VP shunts for the treatment of patients with idiopathic normal pressure hydrocephalus. However, the failure rate was higher for the LP shunt than for the VP shunt in their study. Following our modified surgical procedure significantly decreased the shunt failure rate and the occurrence of complications. The main reason for shunt malfunction of the VP shunt group in our study was obstruction of the proximal catheter by debris in the lateral ventricle, and the debris was found in the side holes of proximal catheter and the valve reservoir. The *Luschka* and *Magendie* foramen may act as filters to decrease CSF debris in the ventricle; therefore, the LP shunt may represent the first choice for treatment of hydrocephalus if we conform with these indications [24].

### Limitations

This study was of a retrospective design and the two groups were heterogeneous patients. Performing a randomized controlled trial was difficult because of the higher surgical costs of the LP shunt. The best way we could perform the study was through the inclusion of the same surgery indications and the matching of the age and sex. However, our surgical procedure design seems to significantly decrease the complication rates. It provides a reference for a surgical technique for neurosurgeons to follow in performing an LP shunt insertion.

## 5. Conclusions

The recent improvements in the quality of the LP shunt device and in proficiency with the procedure have made the LP shunt a safer procedure than the VP shunt for treatment of hydrocephalus. The programmable valve can avoid overdrainage complications and reduce the revision rate. The use of our procedure can circumvent shunt malfunctions due to catheter kinking and bleeding in subarachnoid space while decreasing the revision rates.

## Figures and Tables

**Figure 1 medicina-55-00643-f001:**
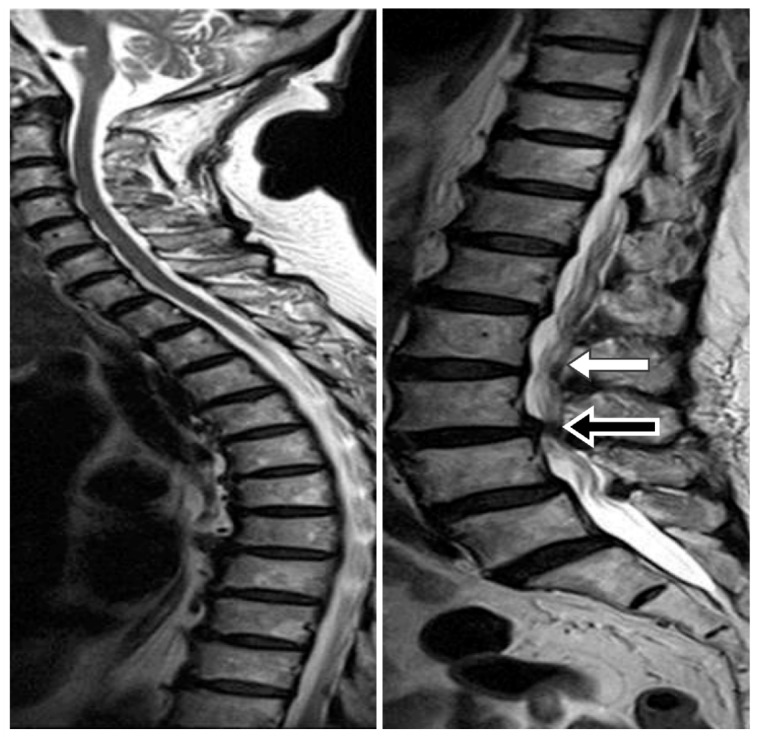
This whole spine MRI showing that we could carry out the tapping step while avoiding the stenosis segment. Taking this case as an example, severe stenosis was noted at L3–4 (black arrow) level, and we were able to perform the puncture easily from L2–3 (white arrow).

**Figure 2 medicina-55-00643-f002:**
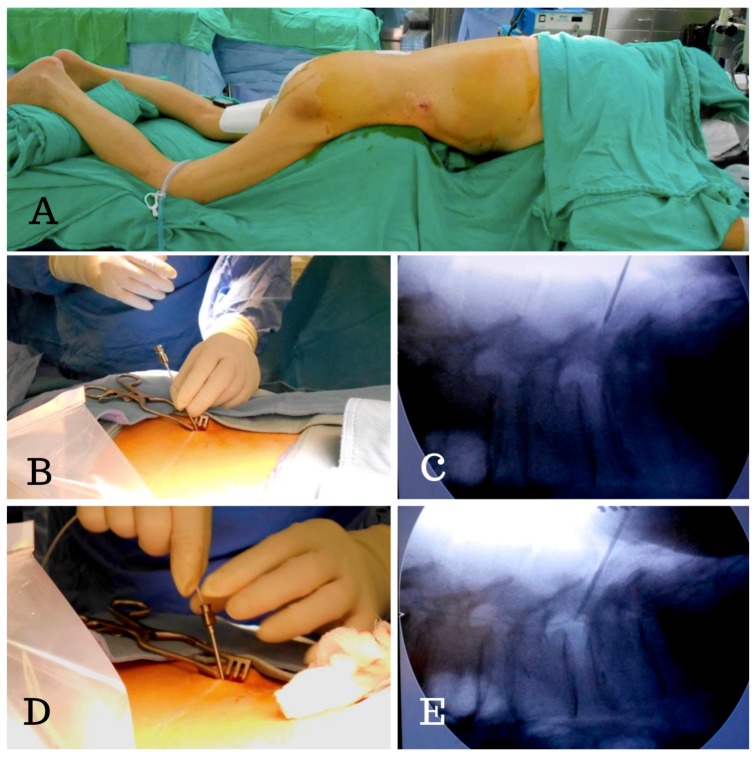
Intraoperative photographs of the placement of the lumboperitoneal (LP) shunt. (**A**) The patient was in a prone position for the first stage of the surgery. (**B**,**C**) The Tuohy needle was inserted into the subarachnoid space at L4–5, (**D**,**E**) followed by insertion of the lumbar catheter into the spinal canal under fluoroscopic guidance. We were able to check the position of the spinal catheter immediately and clearly.

**Figure 3 medicina-55-00643-f003:**
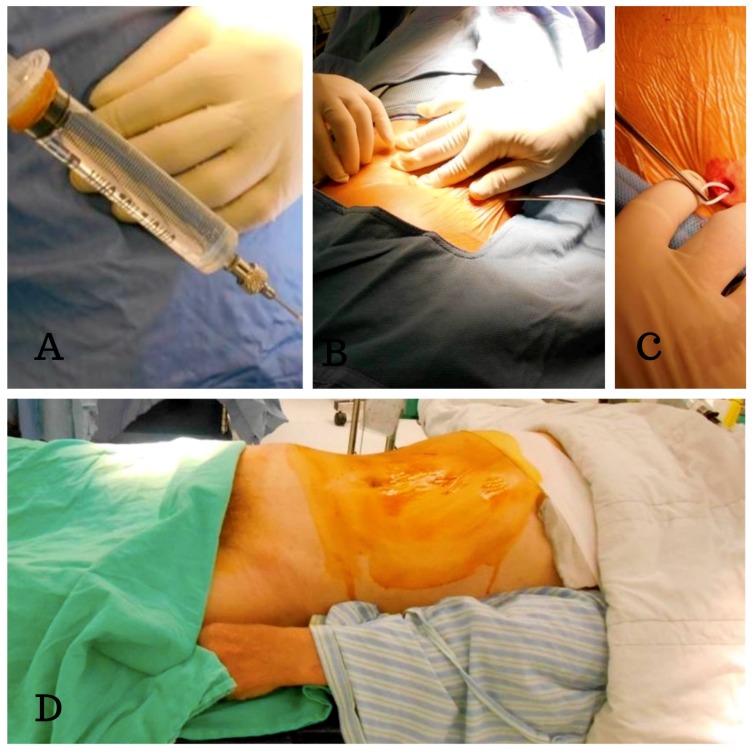
Intraoperative photographs showing placement of the LP shunt. (**A**) The patency of the spinal catheter is checked by gently aspiration. (**B**) We passed the tunneler from the back incision to the flank region. (**C**) The catheter was then placed in a small subcutaneous pocket made at the flank region. (**D**) The patients were in the supine position for the second stage of the surgery (the catheter was connected with a Strata adjustable valve and the abdominal catheter was placed inside the peritoneum).

**Figure 4 medicina-55-00643-f004:**
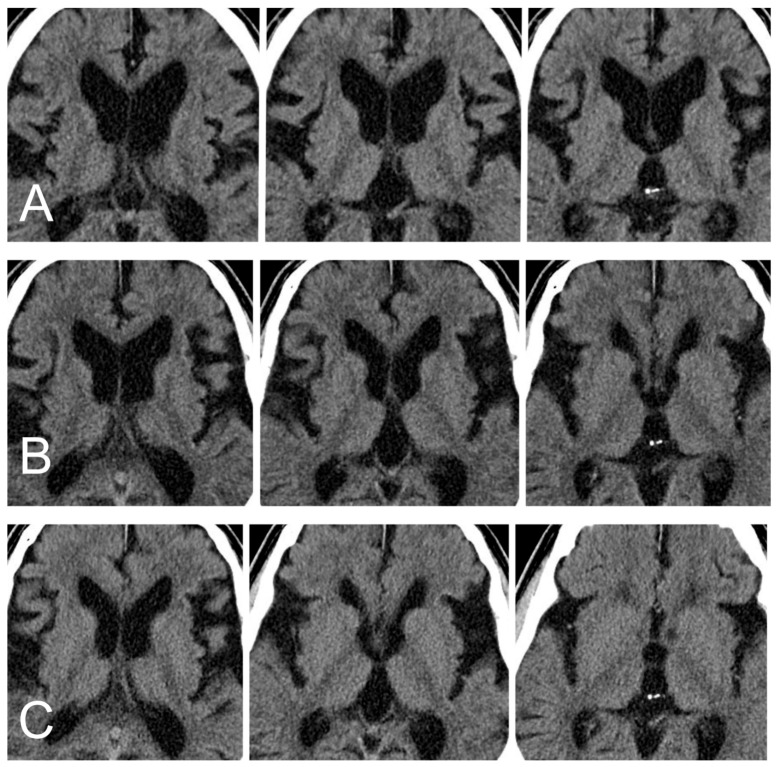
(**A**) The brain CT before the surgery revealed hydrocephalus. The ratio of FH/ID was 0.49. (where FH is the largest width of the frontal horns, and ID is the internal diameter from inner-table to inner-table at this level) (**B**) Brain CT at the third month. The ratio of FH/ID was 0.45. (**C**) Brain CT at the sixth month. The ratio of FH/ID was 0.43.

**Figure 5 medicina-55-00643-f005:**
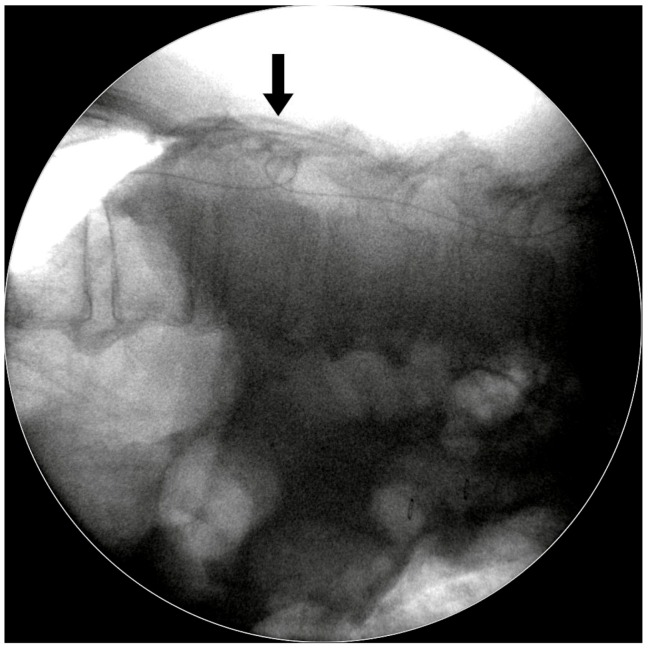
Intraoperative fluoroscopic photograph. The spinal catheter kinked within the spinal canal, but we were able to correct it immediately under fluoroscopic guidance.

**Table 1 medicina-55-00643-t001:** Comparison of the revision rates and complications of the LP shunt and the VP shunt of patients matching with age and sex.

LP Versus VP	LP Group	VP Group	*p* Value ^††^
Total surgery number	96	192	
Revision *	1 (1.0%)	26 (13.5%)	0.001
Shunt malfunction †	0 (0.0%)	14 (7.3%)	N/A
Shunt infection	1 (1.0%)	11 (5.7%)	0.061
Subdural hematoma	1 (1.0%)	5 (2.6%)	0.766

LP: lumboperitoneal shunt; VP: ventriculoperitoneal shunt. *: Revision rate: number of revisions/total number of surgeries †: Shunt malfunctions included shunt obstruction, disconnection, dislocation, and migration. ^††^: Fisher’s exact test.

**Table 2 medicina-55-00643-t002:** Comparison of the revision rate, rate of occurrence of any complication, and infection rate following LP shunt placement in our study versus the VP shunt procedures of Yvonne et al. and Reddy et al.

LP Versus VP	LPS Group Our Study (*n* = 137)	VPS Group Yvonne (*n* = 14,455)	*p* Value	VPS Group Reddy (*n* = 1015)	*p* Value ^†^
Total surgery number	138	N/D		2239	
Revision	1 (0.7%)	1224 (21.6%)	<0.001	1224 (50.2%)	<0.001
Any shunt complication	4 (2.9%)	4192 (29.0%)	<0.001	N/D	N/A
Shunt infection	1 (0.7%)	N/D	N/A	162 (7.2%)	<0.001

*n*: number of patients; N/D: no data; N/A: not available. ^†^: Fisher’s exact test.

**Table 3 medicina-55-00643-t003:** Comparison of the revision rate and other complication rates according to other publications. ACM, Arnold–Chiari I malformation; CSF, cerebrospinal fluid.

LP Versus LP	Ours (*n* = 137)	Aoki (*n* = 204)	*p* Value ^††^	Duthel (*n* = 195)	*p* Value ^††^	Yadav (*n* = 409)	*p* Value ^††^	Bloch (*n* = 33)	*p* Value ^††^
Revision	1 (0.7%)	33 (16.2%)	<0.001	N/D	N/D	44 (11%)	<0.001	9 (17%)	<0.001
Malfunction	0	29 (14%)	N/A	28 (4%)	N/A	32 (7.8%)	N/A	5 (15%)	N/A
Infection	1 (0.7%)	2 (1%)	0.813	10 (5%)	0.028	14 (3.4%)	0.097	2 (6%)	0.038
CSF leakage *	0	0	N/A	0	N/A	12 (2.9%)	N/A	2(6%)	N/A
ACM	0	4 (2%)	N/A	1 (0.5%)	N/A	2 (0.5%)	N/A	0	N/A
Radicular pain	0	10 (5%)	N/A	0	N/A	2 (0.5%)	N/A	0	N/A
Subdural hematoma †	3 (2.2%)	6 (2.9%)	0.679	8 (4%)	0.343	0	N/A	2 (6%)	0.241

*n*: number of patients; N/D: no data; N/A: not available; *: Includes pseudomeningocele; †: Includes subdural hematoma, subdural effusion, and overdrainage; ^††^: Fisher’s exact test.

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
