# Peer review of "Lumboperitoneal Shunt: A New Modified Surgical Technique and a Comparison of the Complications with Ventriculoperitoneal Shunt in a Single Center"

_medicina, 2019, doi:10.3390/medicina55100643_

Round 1

Reviewer 1 Report

Please write detailed figure legends and table legends. Minor changes of your writings are needed.

Reviewer 2 Report

In this MS the authors showed improved surgical method of lumboperitoneal shunt implantation (two-stage procedure) in hydrocephalus. Interestingly the use of programmable valve prevents over-drainage complications and reduce the revision rate. Paper sounds well and agree with the conclusions. There are some minor comments…

Introduction section should be more descriptive. Authors should also provide MRI or CT image of at least one patient after and before treatment, which will help to understand visually as well. Authors are comparing this study with other published studies, like see below lane numbers. It will be great if authors provide more details about other studies likes how they perform and how their study is similar with others.

Lane 138: The same conclusion was made in the series by Aoki et al., Singh et al., and Duthel et al..[5- 7]

Lane 139: We also compared our data for LP shunts with the large number of VP shunts reported by Wu et al.

Lane 143: compared with the infection rate for VP shunts reported by Reddy et al.

Authors should provide detail comparison about surgery procedure between two stage LP vs VP as well.

Reviewer 3 Report

This study addressed the the clinical outcomes of LP shunt in hydrocephalus patients in comparison with traditional VP shunt. I have some suggestions listed below.

Is there any differences of previous abdominal surgery between two groups? In my opinion, previous abdominal operation increased the incidence of adhesion, which might cause distal dysfunction of VP shunt.  VP shunt is considered a training procedure for neurosurgeon residences. Is there any differences between two groups in terms of experiences/year of surgeons? What is the main reason of shunt malfunction in VP shunt group? In my experiences, I placed the shunt into the abdominal cavity by laparoscopy, which provides direct visualization of the placement of tube in the Douglas pouch. By this method, it minimizes the incidence of malfunction of distal shunt.  

Reviewer 4 Report

Although it is an interesting paper on the usefulness of Lumboperitoneal Shunt, however, I have some questions to author.

Q1. In the “Method”, the author should list the matters compared between the two groups.

Q2. I think “patient background” should be described more detail. For example, DM or glucose intolerance considered to be related to the likelihood of infection, was there any difference in prevalence of DM or glucose intolerance between the two groups? was the age different? was the gender ratio the same?, etc.

Q3. Regarding Table1 to Table 3, due to the small sample size of your study, instead of using the standard chi-square test I recommend a continuity adjusted chi-square, and where individual cell size is less than 5 counts, I recommend an Exact chi-square test.

Round 2

Reviewer 3 Report

My comments have been addressed comprehensively.

Reviewer 4 Report

Some part of my questions were not answered, but other part were corrected. Therefore, the modified manuscript of present form is acceptable.